# Analysis and Valuation of the Energy-Efficient Residential Building with Innovative Modular Green Wall Systems

**Elena Korol and Natalia Shushunova ***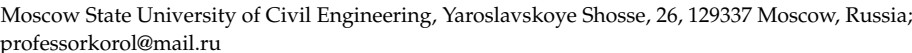

Moscow State University of Civil Engineering, Yaroslavskoye Shosse, 26, 129337 Moscow, Russia;
professorkorol@mail.ru
* Correspondence: nshushun@gmail.com

**Abstract:** The installation of green wall systems on the residential buildings is a complex technological process, the parameters of which vary depending on design solutions, methods of performing work, instrumental and technical support, professional skills of the work performers and many other factors. The authors used the life cycle approach for the assessment of the energy-efficient residential building with integrated greening systems. The aim of the study was to evaluate an energy-efficient residential building with an innovative modular green wall system and to compare it with existing technological solutions. We show that the life cycle approach provides the choice of a decision that is also optimal in conditions of risk, which indicates the effective use of the green wall system. The results of the work are presented by the development of technology with modular green systems, which will expand the practice of technological design, experimental construction and the renovation of buildings, to improve the quality of the urban environment by implementing rational construction and technological solutions and appropriate work methods. This study will be helpful for researchers in green construction to develop their future research studies and for various residential green building owners.

**Keywords:** green wall system; green construction assessment; green building materials; energy-efficient residential building; green building technologies; Life Cycle Assessment

## 1. Introduction

Currently green construction is a significant way to build the strong, healthy and resilient urban systems that we need in the pandemic situation; green urbanization can also be the lifestyle change we need given the post-coronavirus world. There is a significant interest in green construction practices due to a huge number of annual construction activities taking place across the globe.

In the practice of construction design and the implementation of construction processes in the construction of walls, there are various tasks for which the designer has certain information that is implemented in solutions. This information differs in its structure and level of certainty. The main goal of the assessment and multi-purpose selection of technical and organizational-technological solutions for the construction of energy-efficient residential buildings with greening systems is to process the initial information presented in the form of performance indicators contained in the description from the compared options so that it becomes possible to make the final choice of the best option.

The motivation of this study in contrast with the existing solutions is that, by using this methodology, we can evaluate the most cost-effective option for wall structures, and this will be useful in the implementation of the project of innovative modular green wall systems. Moreover, we plan to implement this startup technology at the University.

The valuation of an energy-efficient residential building is based on the model Life Cycle Assessment (LCA). Building Life Cycle Cost (BLCC) is the estimated monetary value of the total costs of owning a residential building, including the costs of construction and

installation works, subsequent maintenance, operation during their service life, repair and the disposal of the elements created as a result of the work buildings or whole buildings.

Using the LCA method to evaluate the sustainability of buildings as a static evaluation method can provide effective tools for process management. Life Cycle Assessment is used as a technical, data-based and holistic method in construction. The studies on LCA of buildings were commonly associated with identifying the most important impact factors for assessment [1,2].

A complex analysis of international experience in the construction and reconstruction of residential buildings with the use of green building technologies revealed a trend toward the integration of the best world practices to create a comfortable living environment. They are based on systemic scientific research and advanced engineering solutions and developments that are technologically related to the operation of residential and commercial buildings, which is carried out in the structural units of the housing and communal complex.

The relevance of the research topic is associated with the need to update the regulatory, technical and technological base in relation to this area based on the results of scientific research and practical experience.

The technological and organizational parameters were studied in the experiments with the thermal assessment of green buildings [3–6]. Most studies focus on the green systems assessment of the vegetation types, physical properties and thermal insulation effects of the green systems [4]. Thus, since 2009, the Construction Technologies Institute of the National Research Council of Italy has been developing a laboratory process for the evaluation and certification of the thermal performances of growing media for green systems [7]. There are many studies carried out using greenery as a mitigation and adaptation strategy to urban heat [8–11] and to measure the green area as a "presence of green" in an urban environment. Green walls are altered due to the impact of human development as a source control measure for urban storm water management, rain water utilization and ecological sewage treatment [12–14].

Green spaces have become key mediators of people's health as people worldwide tend to spend large amount of time in big cities. As modern humans spend more and more of their time indoors, the importance of the interactions between indoor microbiome and human health is becoming more relevant. Recent research studies focus on the biological processes in urban air and the protection of urban populations from toxic substances [15–18].

The arrangement of green wall systems on the buildings is a method to improve the quality of the urban space [16]. The wide range of benefits is associated with green wall systems, including performed technological systems with organizational structures and the reduction in airborne noise and energy cost savings by 40% [19–26]. The effectiveness of green roofs in reducing building energy consumptions in different climatic conditions was investigated [23,24,27]. The benefits of modular construction for high-rise buildings in urban green infrastructure were studied [28–31]. This study examines the rational choice and a comparative assessment of the technological efficiency of an energy-efficient residential building with innovative modular green wall systems based on the valuation of the energy-efficient residential building using the LCA method. There is a unique investigation because it has not been fully explored. The scientific novelty of this work consists in the research and development of rational choices and a comparative assessment of the technological efficiency of a new method of installing a green wall system based on LCA method. The results obtained can be used as a basis for the development of technological regulations and can provide structure in the format of a flow chart for the implementation of organizational and technological design and the construction and reconstruction of buildings with new green wall technology.

## 2. Materials and Methods

The integration of new energy efficient technologies and energy efficient buildings is one of the priority options in the development of modern cities. One of the main

directions of the development of environmental diseases in the cities is the creation of "green islands"—ecological structures on the buildings with greening systems including various types of plants. In addition to the obvious benefits of using these structures in an ecosystem, there is also an energy-saving effect of the city, which is combined with the interaction of building protection and the minimization of heat losses through a counting design for modular systems.

Based on the experience of the construction industry, we can conclude that efficiency indicators obtained as a result of the implementation of projects are worse than those provided in design solutions. In particular, such differences are observed in the development of new technical and organizational-technological solutions during construction. Therefore, to make decisions in conditions of uncertainty, using the LCA method is suitable [32–35]. The LCA method of calculating the costs of energy-efficient residential building is used to compare alternative projects that meet the same requirements for the characteristics of the building, but they differ in ratio capital and operating costs. To justify the introduction in such projects of energy-efficient technologies and materials, a comparison must be performed on the same date estimates.

### 2.1. The Calculation of Building Life Cycle Cost

The total life cycle cost of a residential building includes two categories of costs: one-time costs for commissioning and decommissioning and the recurring costs for the operation and maintenance:

1. One-time commissioning and decommissioning costs include the following:
    1.1. Costs prior to commissioning, including costs for construction and installation works;
    1.2. Disposal costs. Pre-commissioning costs include:
        - The cost of acquiring rights to a land plot;
        - Cost of connection to external engineering (utility) networks, including the following:
            (a) Obtaining technical conditions for connection to external networks;
            (b) The costs associated with this for the reconstruction or modernization of external networks (if their bandwidth or degree of perfection leaves much to be desired);
            (c) Installation works for the construction of networks (gas supply, heat supply, electricity supply, water supply, etc.) from the permitted point of connection to external engineering networks to the building.

The cost of acquiring or leasing land should be included in the original cost estimate if it differs among alternative projects. If they are the same, then they can be ignored when calculating the Building Life Cycle Cost (BLCC). Moreover, the inclusion of the cost of land is necessary, for example, when comparing the costs of the reconstruction of an existing facility and new construction on an acquired land plot.

Construction and installation costs include the following:
- The cost of design;
- The cost of materials and equipment;
- The cost of construction and installation works;
- The costs associated with the diversion of funds for the period of construction (including interest on loans).

At the same time, a detailed estimate of construction costs is not mandatory for a preliminary economic analysis of alternative solutions for building structures and engineering systems. The cost of construction can be determined on the basis of consolidated indicators based on state and non-state standards, unit prices, aggregated indicators of the cost of construction and databases of materials and equipment used.

Disposal costs include the following:

- The cost of demolition work;
- The cost of reusable materials.

The cost of disposing of a facility includes the cost of the demolition of the building minus the cost of reusable materials. The residual value of the system (or component) is calculated at the end of the analysis period, or at the time of its replacement during the analysis period. As a general rule, the residual value of a system that has not yet expired its useful life at the place of installation and operation can be calculated from a linearly proportional distribution of its initial costs.

2. Periodic expenses for the operation and repair during the planned period of operation include:

    2.1. Costs associated with the maintenance of the building;
    2.2. Expenses associated with the acquisition of utility resources from external networks;
    2.3. Costs for current repairs of structures and systems;
    2.4. Costs for overhaul of structures and systems.

Data on the cost of maintenance (operation, maintenance and repair) are obtained from accepted standards or reports from managers of 16 companies, which contain the average cost of ownership and specific operating costs (costs) per unit area (total, residential or usable) depending on the total duration of operation building, region of location and total area of the building.

The costs associated with the acquisition of utility resources include the cost of heat and electricity, water and other utilities. They are obtained on the basis of data on the actual level of consumption and prices that they regulate, seasonal schedules and forecasts of management companies in the housing and communal services sector. In accordance with the principles of green building, the consumption of electricity, heat and water, when designing a building and its enclosing structures, which are interdependent, are estimated for the building as a whole and not for individual building systems or its components. At the initial design stage, data on the amount of energy consumed by the building can be obtained by conducting engineering analysis or using specialized computer programs. When determining energy prices, one should take into account the current and forecast prices of local energy suppliers, the duration of the spring-summer and autumn-winter seasons and demand activity. Water consumption costs are calculated in the same manner as electricity consumption costs. The cost of current and major repairs of structures and systems depends on their service life, physical and functional wear. The starting point for the analysis of future costs associated with the replacement of equipment is the initial cost of this equipment, taking into account the indexation and discounting of the costs of acquiring new equipment.

### 2.2. Impact of the Modular Green Wall Systems on BLCC

The improvement of technologies in green construction is aimed at reducing labor intensity, duration and cost of construction, as well as reducing labor-intensive operations and processes due to the optimal organization of the installation of building structures.

Research on technological processes in green construction and making rational choices for structural and technological solutions allow organizing the rhythmic construction of building structures with greening and covering systems by using modern construction technologies and innovative construction methods and materials [36–38]. New studies demonstrate that recycled wastes and alternative materials can be applied to reduce life-cycle environmental impacts. On the basis of graphical software environments for the three-dimensional modeling of life-cycle management processes of an energy-efficient and environmentally friendly comfortable environment [36], 3 variants of the energy-efficient residential building structures have been developed.

The main principle of LCA methodology is based on reducing the total cost of ownership of a building by a reasonable increase in initial costs at the design stage and the application of energy efficient, environmentally friendly technologies and approaches to green construction, as a result of which a significant reduction in operating costs at an

average of 75% of total costs occurs at the operation stage. Therefore, even if the cost of building an efficient house will be 50% higher than the cost of a standard home, the total cost of ownership for a residential building will be 1.5–2.5 times lower than the cost of living expenses cycle of a standard house through the use of energy-efficient and environmentally friendly technologies that help reduce the cost of maintenance, service and consumed communal resources, which helps reduce the total cost of the building due to the duration of the life of that building. Economic effects on the operation of efficient buildings are expressed in reducing the cost of utilities for residents of these residential buildings.

Thus, the expected economic and social effects are achieved as a result of applying the LCA method of residential building construction, taking into account the cost of the total costs when choosing options for residential building instead of the standard ones; almost 1 billion dollars can be estimated in annual savings in funds only due to the lack of the need to subsidize utility tariffs in energy-efficient apartment buildings.

For the analysis of energy-efficient residential buildings, the definite variants of organizational and technological solutions were developed, represented by Figure 1. The following types of energy-efficient buildings were taken as objects of research:

1.   Residential building with wood cladding (type 1);
2.   Residential building with innovative modular green wall systems (type 2).

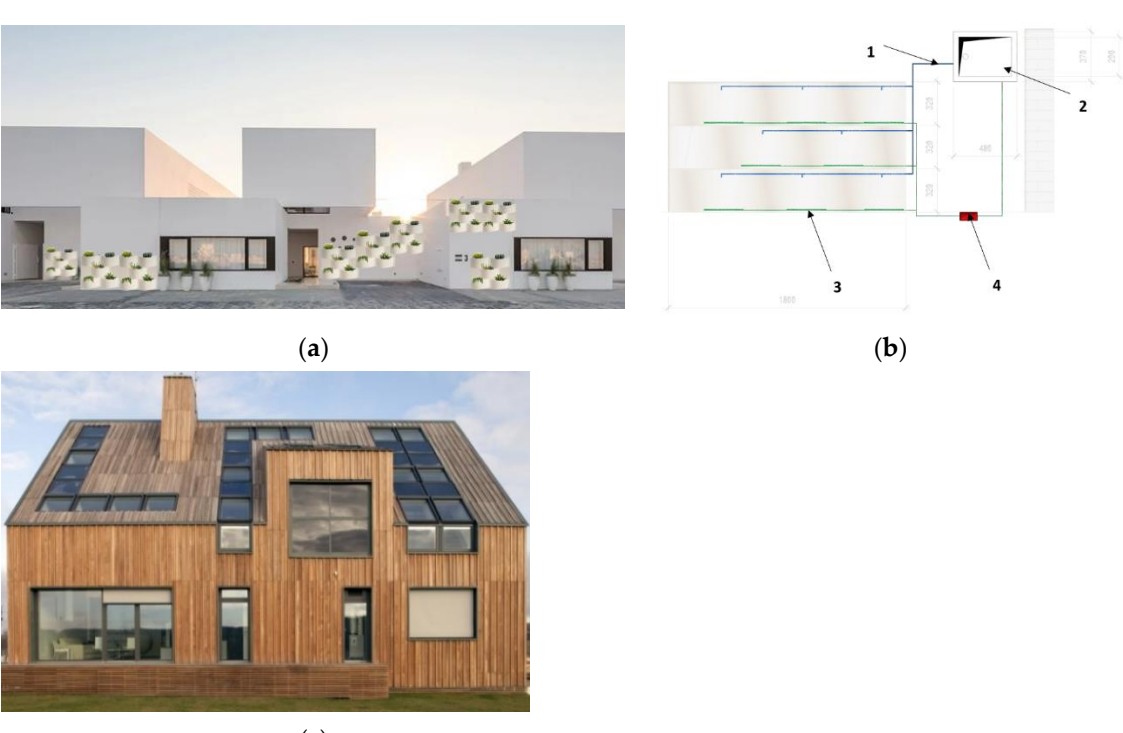

**Figure 1.** Variants of the technological solutions of energy-efficient residential buildings: (**a**) Residential building with innovative modular green wall systems, designed and patented by the authors (3D model); (**b**) sealed to the wall water circulation system; 1—drip irrigation system; 2—water tank; 3—drainage trays; 4—circulation pump with filter; (**c**) residential building with wood cladding.

Green wall system using modular structures, designed and patented by the authors, is an invention that engages living roof modular structures and is a method for providing versatile coverage with the integrated apparatus of energy-efficient devices such as solar heat collectors, semiconductor devices for converting solar energy into electrical energy and special fastenings for attaching irrigation control and that are sealed to the wall water circulation system (Figures 1b and 2b). The impact on seepage is characterized by an economical automatic irrigation system with water recirculation in places of high humidity.

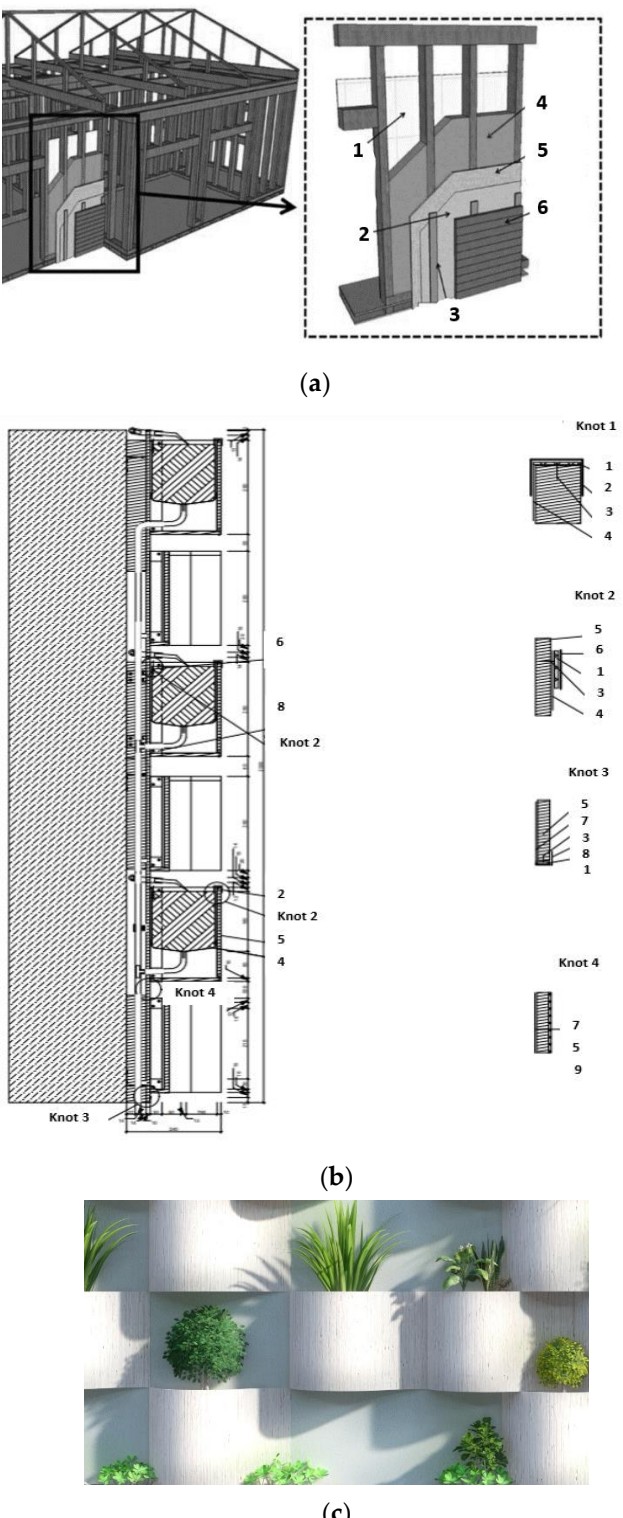

(a)

(b)

(c)

**Figure 2.** Constructive solutions of structures of energy-efficient residential buildings: (**a**) residential building with wood cladding; 1—plasterboard wall; 2—vapor barrier; 3—sheathing; 4—heat-insulating internal material; 5—heat-insulating external material; 6—wood cladding. (**b**) Section view of innovative modular green wall systems, designed and patented by the authors; 1—sealant, 2—profile-WPH-LINE 1212-2000; 3—construction bracket; 4—geotextile; 5—plywood; 6—plastic strip 0.5 × 6.3 mm; 7—vapor barrier; 8—white PVC corner 10 × 10 mm; 9—water-based paint; (**c**) 3D model of innovative modular green wall systems, designed and patented by the authors.

The base of the green wall system using modular structures is represented by a multi-layer monolithic floor wall with a heat-insulating layer of low-heat conductivity and lightweight concrete [37]. Features of constructive and technological solutions for various types of coatings are reflected in the composition of technological operations during the construction of green roof systems. The constructive solutions of structures of energy-efficient residential buildings are shown in Figure 2.

The material from which the green wall structure is built is a moisture-resistant ecoplastic. It does not let water through, assuming that the seams between the wall modules are closed. However, water can drain inside the drop-shaped recesses when the plants are saturated with moisture. To perform this, a structure is provided inside the recess—a geotextile bag that slightly does not reach the very bottom of the recess (Figure 2b).

The integration of new energy efficient technologies for the buildings is one of the priorities in the development of modern cities. The authors have developed an innovative method to improve the environmental situation—the creation of modern green wall modular structures on buildings. In addition to the obvious positive effects from the use of these structures on the city's ecosystem, there is also an energy-saving effect that is based on the simultaneous increase in thermal protections for buildings and the minimization of heat loss through the building envelope by using modular systems. Moreover, the main advantages of the modular technology developed by the authors include the following: reduction in labor intensity by 35% (see Table 1), integration of energy-transforming devices and using the walls as multilevel greening system—urban farming technologies.

**Table 1.** The indicators of the chronometry measurements of the technology of the device of modular green wall systems (per 10 sq.m.).

| Technological Processes and Operations | Duration (in min) of Technological Operations for the Construction of a Modular Green Wall Systems | | | | | | | | | | Cs | Tav | SD |
|---|---|---|---|---|---|---|---|---|---|---|---|---|---|
| | 1 | 2 | 3 | 4 | 5 | 6 | 7 | 8 | 9 | 10 | | | |
| 1. Installation of adjustable supports with a step of no more than 1 m: 1.1. Assembling of the coating for the layout of the supports | 20 | 18 | 17 | 23 | 20 | 18 | 18 | 24 | 19 | 21 | 1.4 | 20 | 2.2 |
| 1.2. Assembling and gluing supports | 22 | 19 | 24 | 20 | 23 | 23 | 22 | 23 | 22 | 22 | 1.3 | 22 | 1.4 |
| 1.3. Adjusting the angle of support | 40 | 46 | 40 | 45 | 41 | 38 | 37 | 38 | 40 | 38 | 1.1 | 40 | 1.7 |
| 1.4 Fixing the special clips | 14 | 10 | 15 | 15 | 14 | 15 | 14 | 15 | 15 | 14 | 1.3 | 14 | 0.8 |
| The duration of technological operations, min (according to claim 1) | | | | | | | | | | | | 96 | |
| 2. Installation of grating 1 × 1 m on the wall: 2.1. Standing the grating on the supports | 26 | 27 | 25 | 26 | 28 | 24 | 26 | 27 | 25 | 26 | 1.1 | 26 | 1.1 |
| 2.2 Fixation of grating | 17 | 17 | 20 | 18 | 18 | 16 | 16 | 18 | 17 | 17 | 1.2 | 17 | 1.8 |
| The duration of technological operations, min (according to claim 2) | | | | | | | | | | | | 43 | |
| 3. Installation of modules for green walls: 3.1. Installation and connection of a group of modules | 22 | 25 | 27 | 28 | 24 | 26 | 26 | 25 | 26 | 27 | 1.1 | 22 | 1.8 |
| 3.2. Fixing a group of modules to the grating on the wall | 20 | 18 | 20 | 22 | 20 | 19 | 21 | 22 | 20 | 18 | 1.2 | 20 | 1.3 |
| 3.3. Filling the group of modules with soil and vegetation | 30 | 35 | 33 | 33 | 34 | 35 | 34 | 32 | 34 | 36 | 1.1 | 30 | 1,9 |
| The duration of technological operations, min (according to claim 3) | | | | | | | | | | | | 72 | |
| Total, Duration of technological operations, min (according to pp. 1–3.) | 296 | 284 | 297 | 300 | 301 | 294 | 295 | 304 | 296 | 295 | - | - | - |
| Total, Average duration of technological operations: | | | | | 211 min. | | | | | | | | |

Cs—stability coefficient of this time series; Tav—the average time for a specific operation; SD—standard deviation.

Chronometry Measurements of the Technological Processes of Installation of the Modular Green Wall Systems

As measurements were taken, each technological process was recorded: date, names of technological processes and operations, the beginning and end of observations and their duration. The timekeeping of technological processes and operations was carried out. The time measurements (indicators) of the duration of technological operations are estimated by the stability coefficient for each time series Cs, which is calculated by the formula following:

$$Cs = Tmax/Tmin, \tag{1}$$

where Tmax is the maximum measurement value in the chronometric series, sec.; Tmin is the minimum measurement value in the chronometric series (Table 1).

The developed green wall solution provides a reduction in labor-intensive processes for buildings due to a collapsible design and the adaptability of the connecting modular elements. The values of the execution time of each process are determined by taking into account the most optimal duration of work and the maximum combination of technological operations. At the same time, the time to complete operations when combining work is reduced by 35%. If 211 min is required during the separate execution of the processes, then the combined operations are performed in 130 min.

### 2.3. Rationale for the Introduction of Green Technologies Coefficients

To designate a construction site that meets the necessary requirements for energy efficiency and environmental friendliness, an energy-efficient building is introduced into the methodology—this is an energy-efficient building designed and built by taking into account the preliminary calculation of the total cost.

The total cost of the life cycle costs of an efficient building takes into account the following:

- For one-time costs—the energy efficiency coefficient, taking into account the costs of the energy efficiency class of the building—Ec;
- For recurring costs—the coefficient of environmental sustainability ("greenness")—Gr.

As the base value of the energy efficiency and "greenness" coefficients of the building, the value corresponding to the minimum required level of energy efficiency class "B" and the minimum level of certification of class "D" according to the system of green building standards was taken. The energy efficiency coefficient—Ec—takes into account the final energy efficiency class of the building in accordance with the energy efficiency requirements for buildings, structures, structures and requirements for the rules for determining the energy efficiency class of residential buildings (see Table 2).

**Table 2.** The value of the coefficient of the energy efficiency class of residential buildings.

| Class Designation | Energy Efficiency Class Name | Deviation Value of the Specific Heat Energy Consumption for Heating, Ventilation and Hot Water Supply of the Building from the Normalized Level, % | Energy Efficiency Coefficient—Ec |
|---|---|---|---|
| A | Highest | less than −45 | 0.55 |
| B++ | Increased | from −36 to −45 inclusive | 0.70 |
| B+ | Increased | from −26 to −35 inclusive | 0.85 |
| B | High | from −11 to −25 inclusive | 1.00 |
| C | Normal | from +5 to −10 inclusive | 1.15 |
| D | Reduced | from +6 to +50 inclusive | 1.30 |
| E | Lower | over +51 | 1.45 |

After establishing the basic level of energy efficiency requirements for buildings and structures, energy efficiency requirements should provide a decrease in indicators characterizing the annual specific value of energy resource consumption in a building and structure at least once every 5 years: from January 2011 (for the period 2011–2015)—by at least

15 percent in relation to the base level; from 1 January 2016 (for the period 2016–2020)—by at least 30 percent in relation to the base level; from 1 January 2020—not less than 40 percent in relation to the base level.

The "greenness" coefficient—Gr—takes into account the final rating of the building according to the standard system of the National Association of Builders STO NOSTROY 2.35.4–2011 «Green construction. Residential and public buildings» [37–39], as shown in Table 3. Since the "greenness" coefficient directly depends on the design of the wall structure, the ranking of the design solution of the wall structure is classified together with the "greenness" coefficient in a single rating system and is subject to evaluation (see ranking mechanism for the best design solution in the Table 3). The ranking of the design solution depends of rating of "greenness" by following criteria: in case where the rating of "greenness" is not certified, the ranking of the design solution has a lower class, and if the rating of "greenness" is a class A certificate, then the ranking of the design solution has a high class; in other cases, the ranking of the design solution (class) is normal.

**Table 3.** The value of the coefficient of "greenness" and the ranking of the design solution of the wall structure of residential buildings.

| Rating of "Greenness" | Ranking of the Design Solution (Class) | Number of Points Scored | Coefficient of "Greenness"—Gr |
|---|---|---|---|
| Not certified | Lower | <260 | 1.15 |
| class D certificate | Normal | 260–339 | 1.00 |
| class C certificate | Normal | 340–419 | 0.85 |
| class B certificate | Normal | 420–516 | 0.70 |
| class A certificate | High | 520–650 | 0.55 |

## 3. Results and Discussion

The choice of the best solution of the energy-efficient residential building is based on the LCA method and the valuation of the energy-efficient residential building by using the rationale of the coefficients of green technologies.

For the purposes of this study, we calculated the calculating the Building Life Cycle Cost (BLCC) for both variants. BLCC is understood as the sum of the current costs of one-time costs and recurring costs for the construction, operation, repair and disposal of a residential building. Then, the formula for calculating the total cost of the life cycle of a building is described as follows:

$$BLCC = C1 \times Ec \times R + Ep \times Gr \times T \times K \times R, \tag{2}$$

where

BLCC—the Building Life Cycle Cost;

C1—the amount of one-time costs for design, construction, commissioning and decommissioning (disposal);

Ep—the sum of periodic expenses during the planned period of operation for resources, maintenance, current and major repairs, consumables, management and wages;

Ec—coefficient of accounting for the energy-efficiency class of the building;

Gr—coefficient of "greenness";

T—the number of periods for repairs and replacement of equipment during the planned service life (life cycle) for each element of the calculation;

K is a correction factor that takes into account seasonality and/or deviation from the standards;

R—discount factor.

All costs are based on prices from the National price book «Housing and civil construction projects».

For residential building with wood cladding, the following description is used.

BLCC = C1 × Ec × R + Ep × Gr × T × K × R = $1050 \times 10^6 \times 0.55 \times 0.6 + 450 \times 10^6 \times 0.6 \times 2 \times 1 \times 0.6 = 671$ millions of conventional units.

For residential building with innovative modular green wall systems designed and patented by the authors, the following description is used.

BLCC = C1 × Ec × R + Ep × Gr × T × K × R = $970 \times 10^6 \times 0.55 \times 0.6 + 407 \times 10^6 \times 0.6 \times 2 \times 1 \times 0.6 = 613$ millions of conventional units.

One-time costs take into account the costs of owners and investors in the initial and final periods of the building's life cycle. They are calculated according to the following formula:

$$C1 = (Cprev + Cinp) + (Cut - Mut), \tag{3}$$

where

C1—the amount of one-time costs for design, construction, commissioning and de-commissioning (disposal);

Cprev—one-time costs before commissioning for the acquisition of land plots, for connection to engineering networks (including the cost of constructing the networks themselves) and building design.

The cost of acquiring a land plot and connecting to utility networks may not be included in the calculation of BLCC if they are the same when comparing alternative projects.

Cinp denotes the one-time costs for input and includes the cost of materials and equipment, the cost of construction, installation, adjustment and other works as well as the costs associated with the diversion of funds for the construction period. At the same time, a detailed estimate of construction costs is not mandatory for a preliminary economic analysis of alternative solutions for building structures and engineering systems. Such estimates are usually not available until the development of the design project, which is a very progressive approach to reducing the cost of structural elements of the future building. The cost of construction can be determined by aggregate indicators in government or commercial prices. These prices are based on indicators of the cost of construction of units of area or construction volume of the building contained in the databases of the materials and equipment used.

(Cut − Mut)—One-time costs for utilization include the cost of recycling materials and structures minus the cost of reuse materials.

For residential buildings with wood cladding, the following formula is used.

C1 = (Cprev + Cinp) + (Cut − Mut) = $(150 \times 10^6 + 264 \times 10^6) + 5 \times 10^6 = 419$ millions of conventional units.

For residential building with innovative modular green wall systems designed and patented by the authors, the following formulat is used.

C1 = (Cprev + Cinp) + (Cut − Mut) = $(150 \times 10^6 + 234 \times 10^6) + 5 \times 10^6 = 389$ millions of conventional units.

The calculation of BLCC can be performed both taking into account inflation and without taking into account inflation—at conditionally constant prices in force on the date of assessment. The discount rate reflects the value of the investor's investments and represents the minimum acceptable profit level for him. For many projects, the rate is calculated based on the requirements of the definite investor.



The discount factor is calculated for each year of the forecast period using the following formula:

$$R = \frac{1}{(1+r)^n} = \frac{1}{(1+0.08)^7} = 0.6 \tag{4}$$

where

$R$—discount factor;

$r$—discount rate (yield) in shares;

$n$—serial number of the year, calculated from the beginning of the forecast period.

The duration of the period includes the terms of design, construction, implementation and provision of services. The review period should be the same for all considered project alternatives. The service life of a residential building begins when all engineering systems of the building are put into operation and the residents are occupied. Typically, a period of 30 years from the date of its commissioning is used to analyze the service life of a building. If discounts are applied, the cost forecasting period may be limited to the period of the next overhaul but not less than 10 years.

## 4. Conclusions

In the conclusions of calculating the cost of the life cycle of a building, according to the LCA methodology and taking into account the requirements of National Building Codes, BLCC was calculated for various types of structures: residential building with wood cladding (type 1) and residential building with innovative modular green wall systems (type 2). This technique takes into account the technological efficiency and energy efficiency of the design. Economic indicators were also obtained—the value of the cost of a building from various structures:

- For residential buildings with wood cladding: 671 million of conventional units;
- For residential buildings with innovative modular green wall systems: 613 millions of conventional units.

In this article, obtaining conclusions about the economic, technological efficiency and energy efficiency of various designs of residential buildings was possible. Thus, the conclusion concludes that the most efficient solution is the design for residential buildings with innovative modular green wall systems, and the least efficient is the design for residential building with wood claddings; the difference in BLCC is about 1 billion rubles. It is also possible to evaluate, in this way and according to the principle of contribution, the value of the innovative technology proposed by the authors—it saves about 58 million conventional units for the building. At the design stage, the use of this methodology will make it possible to determine the most optimal variant, as a result of which the costs of assessing and calculating several options for the installation of structures for energy-efficient buildings will pay off.

**Author Contributions:** This study was designed, directed and coordinated by N.S. and E.K. The authors contributed and proposed the valuation of the energy-efficient residential building with innovative technologies of green wall systems using LCA methodology. E.K. planned and performed the experiment for the determination of technological processes and operations during the installation of green wall systems. N.S. analyzed different building structures such as wood cladding and modular green wall systems on the residential buildings, calculating them in terms of life-cycle cost. The manuscript was written by N.S. and E.K. and commented on by all authors. All authors have read and agreed to the published version of the manuscript.

**Funding:** This research received no external funding.

**Institutional Review Board Statement:** Not applicable.

**Informed Consent Statement:** Informed consent was obtained from all subjects involved in the study.

**Data Availability Statement:** The data presented in this study are available upon request from the corresponding author.

**Conflicts of Interest:** All authors declare no conflict of interest about the representation or interpretation of reported research results.

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
