# Peer review of "Analysis and Valuation of the Energy-Efficient Residential Building with Innovative Modular Green Wall Systems"

_sustainability, doi:10.3390/su14116891_

Round 1

Reviewer 1 Report

This is a very interesting area in which you have presented your findings. I have following questions 

  1. What is the impact on seepage especially in those areas where the humidity is high.
  2. Please plot the results .
  3. References are not in proper format , some of them are even incomplete. 
  4. Provide motivation of the work in contrast with the existing work.
  5. Provide a ranking mechanism for the best design. 

Author Response

Dear Reviewer,

Thank you for your comments,

Please see the corrected file of our Manuscript and comments below,

Kind Regards,

Authors

Reviewer 2 Report

Dear authors, this article, submitted for reviewing to Sustainability, should be structured in a more general way. The results are too specific and have a case study approach. For this, it is necessary to include a wide range of bibliographic references, as those included are insufficient in number and thematic variability. I suggest that the paper be completely revised, proposing broader conclusions, if appropriate. Please note that some English expressions have been written in British English, while in other cases the style is American. Please try to unify the style.

Author Response

(The authors gave the same response as above.)

Reviewer 3 Report

The author is encouraged to review and improve the structure of the document “Analysis and valuation of the energy-efficient residential building with innovative modular green wall systems”, as general recommendations, although, it provides an interesting introduction of the problem, it does not go in depth it its “modular green wall systems”, I understand because it is patented. The research is developed on a comparative analysis of the wood traditional system vs. modular green wall system, but in the same way it does not transmit us the variables of analysis, site conditions, indexes, parameters, duration of data collection; it will be necessary a more consolidated documentary structure, as well as to avoid confusing paragraphs or of opinion not very justified in terms of research, there is information that is repeated frequently, to deepen in the results and conclusions section, and use the correct format in the bibliographic references.

However, the pertinent observations that I will be able to comment on are:

Abstract:

The structure of the abstract should be revised, highlighting more clearly the objective of the research, its methodology and the advantages of the comparative study, emphasizing its results and conclusions.

Line 14-18

The objectives and the instruments used by the author and the use of the "model Life-cycle Assessment" are clear, but it does not describe us the methodology used in the research.

Introduction:

In general, the introduction should be revised, considering the following observations:

Line 27-31: The description of the motivation is worded in a confusing way, currently we have seen the benefit of green building users at the time of CoVid, this background today already scientific should highlight it.

Line 33-40: Revise the structure and wording, expand the paragraph information for better understand it.

Line 42: Describes the LCA as a model as opposed to Line 46 that describes the LCA as a method.

In terms of research there is a difference, which is the applicability that the authors are giving to the presented research.

Revise the wording in these paragraphs to avoid confusion for the reader.

Line 58: The topic is important and of great interest for those of us who are interested in putting research into general practice.

Line 61-71: Review the structure and wording, expand the paragraph as it is confusing and dilates the reading.

Line 81: Clarify where the study is generated, which are its variables and why it is applicable.

Materials and Methods

The author must differentiate whether the document generates an explanatory contribution of how the LCA-BLCC method works, or whether it is a comparative analysis “wood system vs. patented system” and focus such comparisons on analogous procedures or the scientific methodology used in the development of the research, since this only transmit us the approximation of the benefit of its use.

Improve the description of the paragraphs since they are very extensive, identify correctly its primary and secondary sources and the contribution of other disciplines, researchers and investigations that contributed to the process.

Line 92-100: Describe your inputs better as you repeat much of the information described above.

Line 111: Dear author, it will be necessary to be clear and transmit the inputs, variables and specific conditions that served you in your research, since it would be understood that you are describing some steps to follow.

Line 218: It is only in this section that we are shown that we are facing a comparative analysis of the benefits provided by the PATENTED MODULE “quote [37]” compared to traditional models of green walls or adapted to existing buildings.

Line 243: What are the positive effects? dear author, let us visualize the ranges, measures, valuations provided by your model and what are the exceptional conditions of the site for its implementation and that it works properly.

Results and Discussion

Improve the description of the paragraphs since they are very long, it is recommended to the author to separate in paragraphs, in lighter elements of information for a better understanding of the research analysis.

The results should reflect this more substantiated contribution that can be seen in reports and figures the benefit of the applicability of “modular green wall system” versus traditional systems and differentiate those adaptations or regenerations that the paper states.

It is necessary to generate awareness in our community and society of the environmental advantages and energy – economic savings in the use of green walls, but the theoretical model will be only a figure until we do not land with data of the benefits to our society, for this it will be necessary to improve the description of the paragraphs, review the formulas; it is recommended to separate in paragraphs, in lighter elements of less theoretical information and with more practical results for a better understanding of the research analysis.

Conclusions

Describe the conclusions since in the reviewed document there is not this section, this should be focused on each of the findings and the apparent results.

Avoid very general conclusions and that are not justified and documented in the text

Author Response

(The authors gave the same response as above.)

Round 2

Reviewer 1 Report

The authors have not provided any ranking criteria as desired in the first review. Also, the impact of seepage is not addressed properly. 

Author Response

Dear Reviewer,

Thank you for your kind Review,

We corrected our Manuscript,

Please see the new version in attached file,

Warm Regards,

Authors

Reviewer 2 Report

The work can be accepted, in present form, due to the corrections done, especifically those referred to bilbliography.

Author Response

Dear Reviewer,

Thank you very much for your kind Review.

All corrections done, including the bilbliography.

Warm Regards,

Authors